

# High-resolution emission inventory of full-volatility organic compounds from cooking in China during 2015-2021

Zeqi Li[1,2], Shuxiao Wang[1,2], Shengyue Li[1,2], Xiaochun Wang[1,2], Guanghan Huang[1,2], Xing Chang[1,2,3], Lyuyin Huang[1,2], Chengrui Liang[1,2], Yun Zhu[4], Haotian Zheng[1,2], Qian Song[1,2], Qingru Wu[1,2], Fenfen Zhang[1,2], Bin Zhao[1,2]

[1]State Key Joint Laboratory of Environmental Simulation and Pollution Control, School of Environment, Tsinghua University, Beijing, 100084, China
[2]State Environmental Protection Key Laboratory of Sources and Control of Air Pollution Complex, Beijing 100084, China
[3]Laboratory of Transport Pollution Control and Monitoring Technology, Transport Planning and Research Institute, Ministry of Transport, Beijing 100028, China
[4]Guangdong Provincial Key Laboratory of Atmospheric Environment and Pollution Control, College of Environment and Energy, South China University of Technology, Guangzhou, 510006, China

*Correspondence to*: Bin Zhao (bzhao@mail.tsinghua.edu.cn)

**Abstract.** Quantifying the full-volatility organic emissions from cooking sources is important for understanding the causes of organic aerosol pollution. However, existing national cooking emission inventories in China fail to cover the full volatility organics and have large biases in estimating emissions and their spatial distribution. Here, we develop the first emission inventory of full-volatility organics from cooking in China, which covers emissions from individual commercial restaurants as well as residential kitchens and canteens. In our emission estimates, we use cuisine-specific full-volatility emission factors and provincial policy-driven purification facility installation proportion, which allows us to consider the significant impact of diverse dietary preferences and policy changes on China's cooking emissions. The 2021 emissions of volatile organic compounds (VOC), intermediate-volatility organic compounds (IVOCs), semi-volatile organic compounds (SVOCs), and organic compounds with even lower volatility (xLVOC) from cooking in China are 561 (317-891, 95% confidence interval) kt/y, 241 (135-374) kt/yr, 176 (95.8-290) kt/yr, and 13.1 kt/yr, respectively. The IVOC and SVOC emissions from cooking account for 9-21% and 31-62% of the total emissions from all sources in the five most densely populated cities in China. Among all cooking types, commercial cooking dominates the emissions, contributing 54.5%, 66.2%, 68.5% and 46.7% to the VOC, IVOC, SVOC and xLVOC emissions, respectively. The Sichuan-Hunan cuisine contributes the most to total cooking emissions among all commercial cuisines. Residential cooking emissions are also vital, accounting for 22.2%-47.1% of cooking organic emissions across the four volatility ranges, whereas canteens make minor contributions to each volatility range (<10%). In terms of spatial distribution, emission hotspots mainly occur in densely populated areas and regions with oily and spicy dietary preferences. From 2015 to 2021, national organic emissions from cooking increased by 25.2% because of the rapid growth of the catering industry, despite being partly offset by the increased installation of purification facilities. Future control measures need to further promote the purification facilities in commercial restaurants and improve their removal efficiency, as well as reduce emissions from residential cooking. Our dataset and generalizable methodology serve



as valuable resources for evaluating the air quality, climate and health impacts of cooking sources, and help to formulate effective emission control policies. Our national, multi-year, high spatial resolution dataset can be accessed from https://doi.org/10.6084/m9.figshare.23537673 (Li et al., 2023).

## 1 Introduction

Organic compounds are ubiquitous in the atmosphere, exhibiting a continuous volatility distribution spanning both particle and gaseous phases (Robinson et al., 2007). They can be categorized based on saturation vapor concentration ($\log_{10}C^*/(\mu g/m^3)$) into volatile organic compounds (VOC, >6.5), intermediate-volatility organic compounds (IVOCs, 2.5-6.5), semi-volatile organic compounds (SVOCs, 0.5-0.25), and organic compounds with even lower volatility (expressed as xLVOC, <-0.5) (Donahue et al., 2012). All of these organics affect climate, air quality, and human health to varying degrees (An et al., 2023; Zheng et al., 2023a).

Cooking activities are a significant source of organic emissions, as cooking fumes contain many complex organic compounds derived from oil, ingredients, and seasonings (Jin et al., 2021). Source apportionment results based on aerosol mass spectrometry-positive matrix factorization (AMS-PMF method) indicate that cooking organic aerosol (COA) contributes 5%-37% of the total atmospheric organic aerosol (OA) mass concentrations at various urban sites worldwide (Lee et al., 2015; Mohr et al., 2012; Huang et al., 2021; Abdullahi et al., 2013). Additionally, gaseous VOCs, IVOCs and SVOCs emitted from cooking have been identified as crucial precursors of secondary OA (SOA) and O$_3$ (Yuan et al., 2023; Yu et al., 2022; Zhang et al., 2021). Furthermore, I/SVOCs have been reported to produce SOA more efficiently than VOCs and contribute significantly to the OA burden (Zheng et al., 2023b; Jathar et al., 2014). Therefore, quantifying the full-volatility organic emissions from cooking sources is important for understanding the causes of OA pollution and formulating effective policies.

In China, the large and dense population results in a substantial demand for cooking. Furthermore, Chinese cooking is distinguished by its diversity, complexity, and regional variation, setting it apart from cooking in other countries (Lin et al., 2022; Liang et al., 2022). Therefore, cooking emissions in China might exhibit unique significance and characteristics. However, the complexity of cooking emissions in China, including a myriad of distinct emission sources (restaurants serving diverse cuisines, home kitchens and canteens) and thousands of chemical species, poses significant challenges to emission estimation (Lin et al., 2022; Zhao and Zhao, 2018; Liang et al., 2022).

Over recent years, many efforts have been made to quantifying cooking emissions in China. Testing of PM$_{2.5}$ and VOC emission factors (EF) for different cooking cuisines (Lin et al., 2019; Wang et al., 2018a, 2015; Cheng et al., 2016), and surveys on restaurant activity data and purification equipment installations (Jin et al., 2021; Wang et al., 2018a; Li, 2020), have provided necessary data for emission calculations. The use of online oil fume monitoring systems (Yuan et al., 2023) and the use of catering-related point of interest (POI) data (Lin et al., 2022) in the digital map have improved the spatial



resolution of cooking emissions. Small-scale inventories of PM2.5 and VOC cooking emissions have been established in cities or districts such as Beijing, Shanghai and Shunde (Lin et al., 2022; Wang et al., 2018a; Yuan et al., 2023; Qi et al., 2020). At the national scale, a few studies have established cooking emission inventories using relatively simplified methods compared to small-scale inventories (Wang et al., 2018a; Jin et al., 2021; Liang et al., 2022; Cheng et al., 2022), as gathering detailed data over large spatial and temporal scales is difficult. Some national-scale studies have indirectly calculated China's

particle-phase organic carbon (OC) and VOC emissions from cooking by proportionally extrapolating city-scale emissions based on easily obtained statistical data, such as population and catering consumption expenditure (Wang et al., 2018a; Jin et al., 2021). Other studies adopted population or meat consumption as the activity data, and used nationwide per capita EFs and EFs per unit of meat consumption to directly estimate the OC and VOC emissions from cooking nationwide (Cheng et al., 2022; Liang et al., 2022). Both methods above essentially assumed a linear relationship between cooking emissions and

national total activity levels such as population, cooking oil consumption, and meat consumption. Moreover, regarding the pollution control conditions, most studies simply assume that all restaurants are equipped with purification facilities and therefore apply controlled EFs to all restaurants. The only consideration of the restaurants without pollution control is from the study by Jin et al. (2021), which applied purification facility installation proportion (PFIP) survey results in two cities to the whole country.

The above inventories provide a preliminary understanding of national cooking emissions, but they still have major shortcomings and considerable uncertainties. Firstly, the existing national cooking inventories fail to cover the full-volatility organics. They primarily consider gaseous (VOC) and particle-phase primary OA (or related OC or PM2.5), but miss the important gaseous I/SVOC emissions, which may lead to significant underestimation of SOA formation. Besides, Chang et al. (2022) have developed a full-volatility emission inventory for China for most emission sources, but the cooking source

was missing from the inventory, possibly due to the lack of EFs, which hinders an accurate understanding of OA sources. Benefiting from advanced measurement techniques, full-volatility organic EFs have recently been measured for different cooking sources (Yu et al., 2022; Song et al., 2022; Huang, 2023). This makes it possible to establish an unprecedented full-volatility organic cooking emission inventory, but such efforts have not yet been made.

Moreover, previous national inventories suffer from significant biases in the estimates of emissions and spatial distributions.

The statistical data currently used for emission calculations hardly reflect the complex cooking activities in millions of commercial restaurants, countless home kitchens and canteens. Meanwhile, the relationship between emissions and national statistics is not simply linear, because different regions have vastly different dietary habits, cooking styles, and cooking pollution control policies, leading to large differences in EFs and PFIPs (Jin et al., 2021; Lin et al., 2022). These issues introduce large uncertainties in emission estimation. Most importantly, the aforementioned methods cannot accurately

describe the spatial distribution of cooking emissions, which is crucial due to the strong linkage between the location of cooking emissions and human living environments, potentially posing significant health risks (Lin et al., 2022; Wang et al., 2018a).



In this study, we develop the first inventory of full-volatility organic emissions from cooking sources in China, encompassing high-resolution emissions from each individual commercial restaurant, as well as family kitchens and canteens, during 2015-2021. We estimate the emissions using cuisine-specific EFs and dynamically changing PFIPs driven by provincial-level control policies. Further, we analyze the sources, regional variations, and temporal trends of full-volatility cooking emissions in China. We also quantify the contribution of key drivers to emission changes and provide recommendations for future control strategies.

## 2 Methodology and data

We use the emission-factor method to estimate organic emissions from three types of cooking activities, namely commercial cooking, residential cooking, and canteen cooking, essentially covering all dietary sources for people (Liang et al., 2022). Notably, we focus solely on cooking fume emissions, excluding emissions from cooking fuels, which have been included in the domestic combustion source in our previous full-volatility inventory (Chang et al., 2022). We use different calculation methods for the three sources according to their characteristics and data availability. The most important commercial cooking is treated as a point source, with detailed cuisine types and geographic coordinates used to estimate the emissions of each individual restaurant and pinpoint its location. Residential and canteen cooking are estimated by province (Section 2.1). The data used to calculate emissions are derived from multiple sources or from our estimates (Section 2.2). The emissions are allocated to spatial grids using the exact locations of commercial cooking and using spatial proxies for residential cooking and canteen cooking (Section 2.3). Finally, we analyze the uncertainty of the inventory (Section 2.4) and quantify the contribution of different drivers to emission changes through sensitivity analysis (Section 2.5).

### 2.1 Emission calculation method

### 2.1.1 Emissions of commercial cooking

The commercial catering industry in China is varied and complex, with its emissions influenced by the diversity of cuisines and regional pollution control regulations (Lin et al., 2021; Song et al., 2022; Amouei et al., 2017; Lin et al., 2022). Our emission calculations, based on the point sources of cuisine-specific restaurants and installation status of purification facilities driven by policy changes, fully consider these influencing factors. We capture the geographic location of nearly all commercial restaurants (up to 7.70 million) nationwide and identify their cuisine types. For each restaurant, we calculate its activity data, i.e., the volume of cooking fumes, and adopt the corresponding full-volatility EFs depending on its cuisine type. Previous studies often simplistically apply the controlled EFs to all restaurants when calculating cooking emissions (Liang et al., 2022; Wang et al., 2018a; Lin et al., 2022), overlooking that over 30% of restaurants do not have fume purification facilities (Jin et al., 2021). Here, we estimate the PFIP in each province to consider the excess emissions from these restaurants without purification facilities. Since it is challenging to know the installation situation of purification facilities for each restaurant, we use the provincial-level PFIP to weigh the controlled EFs and uncontrolled EFs, forming a



comprehensive EF for restaurants of each cuisine within each province, applicable to all restaurants in that category. In this
way, we can obtain an overall emission for each type of restaurant that is closer to reality, as compared to the previous
method of applying controlled EFs to all restaurants. The emissions from commercial cooking are estimated as shown in
Eq.(1):

$$E_{c,p,v} = \sum_{n=1}^{N_{c,p}} A_c(n) \left[ EF_{c,v} y_p + EF'_{c,v}(1 - y_p) \right] \tag{1}$$

where the subscript $c$ represents the cuisine types; $p$ represents the provinces; $v$ represents the volatility bin, with saturation
vapor concentration ($C^*$) varying from $10^{-2}$ to $10^7$ μg/m³ at 300 K based on the framework of Chang et al. (2022). $N_{c,p}$ is the
number of restaurants of each cuisine in each province; $A_c(n)$ is the annual fume gas volume of the $n^{th}$ restaurant, m³/y. $EF_{c,v}$
and $EF'_{c,v}$ are the controlled and uncontrolled organic EF for each cuisine in each volatility bin, respectively, μg/m³. $y_p$ is the
PFIP for each province. Details on data acquisition are provided in Section 2.2.

**2.1.2 Emissions of residential cooking**

Residential cooking refers to meal preparation at home for individuals or families, where most dishes are cooked with
common oil, ingredients, and seasonings using simple cooking methods (Liang et al., 2022). Moreover, the fumes emitted
during cooking in home kitchens are generally expelled outdoors through range hoods, exhaust fans, or natural ventilation
(Qi et al., 2020). The main functions of range hoods and exhaust fans are to reduce the concentration of pollutants indoors,
but they have almost no removal effect on the organics in the fumes. Therefore, we use a uniform uncontrolled EF for
residential cooking. Meanwhile, we use official statistics (National Bureau of Statistics of China, 2022c) of household edible
oil consumption as activity data due to its minimal uncertainty and strong correlation with cooking emissions (Jin et al.,
2021). The emission from residential cooking is calculated by Eq.(2):

$$E_{p,v} = A_p \times EF_v \tag{2}$$

**2.1.3 Emissions of canteen cooking**

Canteen cooking, often featuring simple, low-oil meals with fixed ingredients, caters to students and employees in
enterprises and institutions (Liang et al., 2022). Given the consistent diners and dining regularity, we calculate canteen
cooking emissions based on the number of meals served and also a uniform EF:

$$E_{p,v} = A_p \times \left[ EF_v \times y'_p + EF'_v(1 - y'_p) \right] \tag{3}$$

where $A_p$ is the annual total number of meals served in canteens in each province, meals/y; $EF_v$ and $EF'_v$ are the organic
emission per meal in each volatility bin after and before pollution control, in g/meal; $y'_p$ is the PFIP for canteens in each
province.



## 2.2 Data acquisition and processing

### 2.2.1 Activity data

For commercial cooking, we capture nearly all restaurants nationwide and estimate their annual cooking fume volumes ($A_c$) by gathering catering-related POI data on digital maps and collecting multi-source statistical data. We extract POI data from Amap, a digital map platform, via a web application programming interface (API), following the method of Wu et al. (2021). The information provided by POI data includes the name, labels, longitude and latitude of millions of catering service venues

across China. It offers broad and timely coverage with high spatial resolution, outperforming the population statistics, cooking oil consumption and meat consumption used in most previous inventory calculations (Li et al., 2019).

Identifying the cuisine type of each restaurant is crucial for mapping it to the corresponding EFs and accurately estimating its emissions. The three-level labels in the POI data assist in categorizing restaurants, but they may not be precise enough, as over 60% of restaurants are simply labeled as Chinese food restaurants. Therefore, we classify the restaurant cuisine by

searching their names and labels for specific terms related to certain cuisines (see Table S1). The specific terms are obtained through word frequency analysis using the 'jiebaR' package in the R statistical framework version 4.0.3 (R-4.0.3). The specific classification method is described in Text S1. We finally assign all restaurants to the nine cuisine types (see Table S1 for their characteristics) supported by full-volatility organic EFs: home-style cuisine, Chinese fast food and snacks, hotpot, barbecue, Sichuan-Hunan cuisine, Guangdong-Fujian cuisine, Jiangsu-Zhejiang cuisine, other Chinese cuisines and non-

Chinese cuisines. Here, we've excluded catering services without fume emissions, such as tea houses and coffee houses.

Next, we estimate the $A_c$ of restaurants. According to national standards, Restaurants can be divided into three sizes, large, medium, and small (MEE, 2001), with different activity levels for each size. However, due to the lack of detailed statistics on the size of each restaurant, we can only estimate a scale-weighted average $A_c$ for each cuisine type based on existing data following the equation below, which is used as the activity level of each restaurant belonging to that cuisine type:

$$A_c(n) = \sum_{s=1}^{3} x_{c,s} N_{c,s} Q_s T_s \tag{4}$$

where the subscript $c$ represents the cuisine types; s represents the restaurant scales. $x_{s,c}$ is the proportion of restaurants of different scales; $N_{c,s}$ is the average number of stoves in a restaurant; $Q_s$ is the cooking fumes discharge rate of each stove, m³/h; $T_s$ is the annual total operating time of restaurants, h/y. The values of the above parameters are derived from multiple surveys and literature (Lin et al., 2022; Wang et al., 2018a, b; Yuan et al., 2023), as detailed in Table S2. It is noted that the values of $x_{s,c}$ and $N_{c,s}$ depend on cuisine type.

The activity data for residential cooking, the annual household edible oil consumption, is calculated by multiplying per capita oil consumption with the resident population, derived from the official statistical yearbooks (National Bureau of



Statistics of China, 2022b, c). Besides, the activity data of canteens is the annual total number of meals provided by canteens, which is calculated by Eq.(5):

$$A_p = \sum_{l=1}^{6} n_{p,l} D_l m_l z_l \tag{5}$$

where the subscript $l$ represents six different populations, including preschool and kindergarten students, primary school
students, junior high school students, high school students, undergraduate and graduate students, and employees of state-owned and collective enterprises and institutions. $n_{p,l}$ is the number of students or employees of the six populations in each province. $D_l$ is the average annual number of days in school or at work for various populations; $m_l$ is the average number of meals per day in the canteen for various populations; $z_l$ is the proportion of people dining in canteens for each type of diner. The values of the above parameters are determined by official statistics (National Bureau of Statistics of China, 2022c, a)
and empirical estimation, which are described in detail in Table S3.

**2.2.2 Controlled and uncontrolled full-volatility emission factors**

Our main advances over traditional cooking inventories are to cover full-volatility organic emissions, and to consider differences in regional cuisines and variations in the installation of purification facilities. To accomplish this, we provide a set of controlled and uncontrolled full-volatility organic EFs for nine different commercial cuisines as well as residential and
canteen cooking.

The full-volatility EFs for various commercial cuisines are mainly derived from full-volatility measurements of gaseous and particle-phase organics (Huang, 2023; Song et al., 2022; Yu et al., 2022; Song et al., 2023), and are supplemented by other cooking emission test results recorded in the literature (Xu et al., 2023; Wang et al., 2018a; Cheng et al., 2016; Huang et al., 2020; Sun et al., 2022; Jiang et al., 2021; He et al., 2020; Tong, 2019; Xu et al., 2017; Lin et al., 2019; Li et al., 2020; Zhang
et al., 2016; Shu et al., 2014; Wang, 2013; Li et al., 2021; Lin et al., 2014; Zhao et al., 2007; He et al., 2004; Wang et al., 2018b; Pei et al., 2016). As most commercial restaurants have installed purification facilities, the existing full-volatility EFs of commercial cooking are all measured after pollution control. Therefore, we first obtain a set of controlled full-volatility EFs for all cuisines. Existing full-volatility tests have covered commercial restaurants featuring home-style cuisine, hotpot, barbecue, Sichuan-Hunan cuisine, other Chinese cuisines and non-Chinese cuisines (Huang, 2023; Song et al., 2023), so we
can obtain the controlled full-volatility EFs for these cuisines. However, not all of the nine cuisines mentioned in 2.2.1 have full-volatility tests for both gaseous and particle-phase organics. Therefore, for the commercial cuisines lacking full-volatility testing of gaseous or particle-phase organics, we estimate and supplement the missing full-volatility EFs based on the literature. Specifically, for cuisines lacking the gaseous full-volatility EF, we adopt the average VOC EFs from corresponding cuisines in previous studies (Xu et al., 2023; Wang et al., 2018a; Cheng et al., 2016; Huang et al., 2020; Sun
et al., 2022; Jiang et al., 2021; He et al., 2020; Tong, 2019; Xu et al., 2017) to determine organic emissions within the VOC range ($\log_{10} C^*/(\mu g/m^3) \geq 7$), and then proportionally estimate EFs of gaseous organics in other volatility bins based on the



volatility distribution of gaseous organics emitted from similar cuisines; for cuisines lacking the particle-phase full-volatility EF, we use the average primary OA (POA) EFs from previous studies (Lin et al., 2019; Zhang et al., 2016; Li et al., 2020; Shu et al., 2014; Wang, 2013; Li et al., 2021; Lin et al., 2014; Zhao et al., 2007; He et al., 2004; Wang et al., 2018b, 2015;

Pei et al., 2016) as the total particle-phase organic EFs, and distribute the total particle-phase EFs into each volatility bins following the volatility distribution of particle-phase organics emitted from similar cuisines. Detailed data sources and methods used to estimate all full-volatility EFs are described in Table S4-5. POA EFs were rarely given directly in previous studies, but they can be calculated from $PM_{2.5}$ EFs. Given that the majority of particles emitted from cooking activities are $PM_{2.5}$ (~94.0% (Buonanno et al., 2009)) and the particles consist primarily of organics (69.1%~84% (Pei et al., 2016; Zhao

et al., 2007), median at 76.6%), we assume that the POA EFs equates to 81.5% (76.6%/94.0%) of the corresponding $PM_{2.5}$ EFs. After determining both gaseous and particle-phase full-volatility EFs, the total full-volatility EFs for each type of cuisine are computed as the sum of these two components.

Next, we estimate the uncontrolled EFs for each commercial cuisine based on the controlled EF and removal efficiency of the corresponding cuisine. In the absence of removal efficiencies for I/SVOCs, we assume that the removal efficiencies for

gaseous and particle-phase organics are equal to those for $PM_{2.5}$ and VOCs, respectively, similar to our previous study's approach (Chang et al., 2022). Since most (>90%) purification devices in commercial restaurants are electrostatic fume purifiers (Liang et al., 2022), we adopt a uniform removal efficiency for the purification devices in all restaurants of the same cuisine type. The removal efficiencies of $PM_{2.5}$ and VOCs for each cuisine type are determined by comparing uncontrolled and controlled EFs in numerous previous studies, as shown in Table S6 and Fig. S1.

The full-volatility EFs for residential cooking also come from full-volatility measurements of gaseous and particle-phase organics (Song et al., 2023, 2022; Huang, 2023), as specifically described in Table S4. Since the emissions from residential cooking are generally exhausted through range hoods or exhaust fans without any purification (Liang et al., 2022; Qi et al., 2020), and the existing full-volatility measurements are also conducted under uncontrolled conditions (Song et al., 2023, 2022; Huang, 2023), we only need to obtain uncontrolled EFs for residential cooking. The original test data is provided in

the form of emission rate (μg/min). To match the activity data used for emission calculations, we convert the EFs for residential cooking into the emissions per unit consumption of cooking oil (g/kg oil), according to the method detailed in Text S2.

The controlled and uncontrolled full-volatility EFs for canteen cooking are determined using the same method as commercial cooking, based on a series of emission tests listed in Table S4 (Huang, 2023; Liang et al., 2022; Wang et al., 2018a; Zhao et

al., 2020). To match the activity data used for emission calculations, we convert the EFs for canteen cooking into the emissions per meal (g/meal), according to the method detailed in Text S2.



### 2.2.3 Fume purification facility installation proportion

The treatment of the PFIP has always been a weakness in previous cooking emission inventories. Most studies simplistically assume that all commercial restaurants have fume purification facilities (Liang et al., 2022; Wang et al., 2018a; Cheng et al.,

2022). While a few studies obtained citywide PFIP through door-to-door restaurant surveys (Jin et al., 2021; Li, 2020), the time and labor-intensive nature of the surveys constrains their spatial and temporal coverage. So far, PFIP survey results are only available for three cities. To overcome this limitation, we gather multi-year policies related to catering emission control in each province, and extrapolate the PFIPs from specific years in three cities to multiple years (2015-2021) in China's 31 provinces based on the assumption that the installation proportions correlate closely with the stringency of local regulations.

Specifically, we first construct the relationship between the stringency of pollution control policies and PFIP based on the situation of the three cities (Heze, Linfen, Nanjing) with detailed PFIP survey data (Jin et al., 2021; Li, 2020), as shown in Table 1. Notably, the pollution control scenarios of these three cities comprise include both cases of strong and weak regulatory forces. When combined with the optimal scenario (PFIP reaching 100%), they essentially cover the various pollution control statuses across different regions nationwide.

**Table 1: Grading standards for provincial catering emission control stringency and the PFIPs corresponding to each control stringency level.**

| level | control stringency | detail description | provincial PFIP for restaurants of different scales | | |
|---|---|---|---|---|---|
| | | | large | medium | small |
| A | full implementation | the target year when local standards or policies explicitly require all restaurants to install purification facilities and subsequent time | 100% | 100% | 100% |
| A- | transition to full implementation | the period between the year of the release of policies explicitly requiring 100% PFIP and the target year of meeting these requirements. | the PFIPs linearly increase from the B-level PFIPs to A-level PFIPs | | |
| B | complete regulation | the third year after the release of a control policy that covers all restaurants in the province | 82.8% | 72.8% | 59.9% |
| B- | transition to complete regulation | the first and second years after the release of a control policy that covers all restaurants in the province | the PFIPs linearly increase from the C-level PFIPs to B-level PFIPs | | |
| C | partial regulation | the state when only certain areas or catering services are controlled (for example, the regulation only in the provincial capital city, barbecue or large restaurants) | 64.0% | 59.0% | 41.0% |

We divided the stringency of the control policies into different levels. Among the three cities, Heze is classified as level C (partial regulation). In the year of the PFIP survey in Heze, Heze's catering control policy only included spot checks on

restaurants in busy food streets. Linfen and Nanjing are classified as level B (complete regulation). At the time of the survey, their policy state was such that citywide catering industry emission control policies had been implemented for three years. The PFIP survey results of the three cities are used for PFIPs of level B and level C. However, although the introduction of



citywide emission control policies can increase the PFIPs,  it does not guarantee that all restaurants will adopt the requisite purification facilities, as evidenced by the PFIPs for level B. Further, the optimal level A (full implementation), represents

the target year wherein local standards or policies explicitly require all restaurants to install the purification facility. It's easy to know that the corresponding PFIP for level A is 100%. Since the policy may not be implemented immediately after it is issued, we also consider a transition period, i.e., the status corresponding to A- and B- in Table 1. The transition period from the introduction of a citywide control policy to the achievement of level B is considered to be three years, referring to the situation of the two surveyed cities (Jin et al., 2021; Li, 2020). The transition period from the announcement of the full

implementation policy to the achievement of level A depends on the target time specified in the policy. During the transition period, PFIP is considered to increase linearly. Besides, if there is no new policy issued, the PFIPs will remain unchanged.

Next, we collate key policy milestones and implementation transition periods of catering pollution control policies in each province to determine the level of control stringency, thus obtaining the corresponding PFIP, based on Table 1. Note that the PFIPs of restaurants of different scales may vary. Therefore, for commercial restaurants, the PFIPs are taken as a weighted

average according to the scale proportion of different scales (see Table S2). As for canteens, since they typically cater to large numbers of students and employees, we approximate the canteen's PFIPs to be consistent with that of large-scale restaurants in the same regions.

### 2.3 Grid allocation

To examine regional emission variations, pinpoint hotspots, and assess emission impacts on air quality, a grid inventory for

cooking emissions is necessary. Here we allocate cooking emissions in China into grids with a 27 km×27 km spatial resolution, utilizing catering-related POI data and the nationwide population density data with a spatial resolution of 1 km×1 km. Gridded data sets with higher resolution of up to 1 km×1 km can be prepared with the same method. For commercial catering, we have developed an emission inventory with point source accuracy. We simply aggregate the emissions of restaurants located within each grid to obtain the gridded inventory, with calculations performed in R-4.0.3 to handle the

massive computational load. Besides, emissions for canteens and residences are allocated to the county level based on tertiary industry gross domestic product (GDP) (National Bureau of Statistics, 2022c), and then distributed to grids per population density.

### 2.4 Uncertainty analysis

We employ Monte Carlo simulations to estimate the uncertainties in emissions by considering the probabilistic distributions

of key parameters. Our approach to quantifying the uncertainties of the parameters is generally consistent with Chang et al. (2022). Specifically, the activity data is assumed to follow a normal distribution with coefficients of variation (CVs) between 5-50%, based on the standard proposed by Li (2017). The EFs are assumed to fit the log-normal distribution with CV values, according to Wei (2009). Notably, the uncertainty of the full-volatility EFs estimated from VOC and POA EFs would be 30%



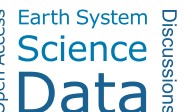

higher than that of full-volatility EFs obtained from direct testing. Furthermore, the unit conversion would add an additional
20% to the uncertainty. Additionally, the CVs for purification efficiencies and installation proportions of fume purification facilities are assumed as 20% and 30%, respectively (Li, 2017). Then, we conduct 10,000 iterations of the simulation, which yields results in the form of statistical distributions. This enables us to ascertain the uncertainty ranges for emissions from various sources at a 95% confidence level.

## 2.5 Sensitivity simulations

We conduct a series of sensitivity analyses to explore the factors driving changes in cooking emissions during 2015-2021. Direct influencing factors of cooking emission changes include variations in the catering industry (specifically, the change in the total number of restaurants and the cuisine distribution), pollution control enhancement, changes in edible oil use and changes in canteen diners. These factors are also indirectly affected by changes in the external environment, such as economic growth, population migration, and the COVID-19 pandemic. We use a brute-force method to quantify the annual
impact of the four factors from 2015 to 2021, in which we sequentially adjust the values of an individual factor to their next year's value. The emission difference pre and post-adjustment are seen as that factor's contribution to emission change for that year.

## 3 Result and discussion

The database provided in this study includes emission calculation parameters and emission inventories. We provide a set of
full-volatility EFs applicable nationwide, high-resolution activity data, and dynamically changing PFIPs (Section 3.1), which are beneficial for calculating emission inventories at different periods and regions. Concurrently, the analysis of emission characteristics, including volatility distribution (Section 3.2), spatial distributions (Section 3.3), and temporal evolution trends (Section 3.4), provides comprehensive insights into cooking emissions.

### 3.1 Full-volatility emission factors, cooking activity data, and purification facility installation proportion

In this study, we present a set of controlled and uncontrolled full-volatility organic EFs for cuisine-specific commercial cooking, as well as residential and canteen cooking (Table 2 and Fig. 1). Commercial restaurants and canteen may or may not have purification facilities for the pollutant removal, while fumes from home kitchens are usually expelled through range hoods or exhaust fans without purification (Liang et al., 2022; Qi et al., 2020). Therefore, we provide both controlled and uncontrolled EFs for commercial cooking and canteen cooking, and uncontrolled EFs for residential cooking. According to
Fig. 1, the significant variance in EFs across the nine commercial cuisines demonstrates the necessity of distinguishing among cuisines when quantifying commercial cooking emissions. Sichuan-Hunan cuisine exhibits the highest controlled EF (11498 μg/m³) among the nine commercial cuisines, attributed to its high oil consumption and the extensive use of spicy seasonings such as chili and pepper. Barbecue ranks second in controlled EF value (9430 μg/m³), largely due to the high heat



levels that facilitate complex chemical reactions and the extensive use of seasonings. Comparatively, home-style cuisine and
non-Chinese cuisines show the lowest emissions (1555 μg/m³ and 1673 μg/m³), probably because of their less frequent usage
of high-emission cooking methods such as frying and grilling compared to other local specialty cuisines. Besides, residential
cooking uncontrolled EF is 20.3 g/kg oil, and canteen cooking controlled EF is 0.648 g/meals. The volatility distribution of
EFs across all cooking sources is similar. VOCs dominate the cooking organic emissions (~55%), followed by SVOCs (17-
33%) and IVOCs (11-36%), while xLVOCs are negligible (<2%).

**Table 2: Controlled and uncontrolled full-volatility EFs for different cooking sources.**

| type of source | | $\log_{10}C^*$ (μg/m³) | | | | | | | | | |
|---|---|---|---|---|---|---|---|---|---|---|---|
| | | ≤-2 | -1 | 0 | 1 | 2 | 3 | 4 | 5 | 6 | ≥7 |
| **controlled EFs** | | | | | | | | | | | |
| commercial cooking (μg/m³) | home-style cuisine | 8.84 | 12.9 | 62.0 | 393 | 102 | 102 | 61.4 | 59.5 | 199 | 554 |
| | Chinese fast food and snacks | 8.22 | 11.6 | 62.0 | 557 | 65.2 | 65.3 | 84.1 | 54.5 | 309 | 1824 |
| | hotpot | 20.5 | 14.3 | 108 | 195 | 107 | 142 | 106 | 123 | 524 | 1939 |
| | barbecue | 81.1 | 26.8 | 196 | 201 | 668 | 336 | 397 | 314 | 1798 | 5412 |
| | Sichuan-Hunan cuisine | 76.4 | 61.2 | 423 | 1165 | 743 | 962 | 402 | 473 | 1907 | 5285 |
| | Cantonese-Fujian cuisine | 12.3 | 12.2 | 64.9 | 460 | 111 | 117 | 83.0 | 67.8 | 285 | 1377 |
| | Jiangsu-Zhejiang cuisine | 12.8 | 12.7 | 67.7 | 482 | 115 | 121 | 86.7 | 70.5 | 298 | 1446 |
| | other Chinese cuisines | 15.5 | 17.8 | 94.2 | 621 | 144 | 146 | 138 | 111 | 425 | 2169 |
| | non-Chinese cuisines | 2.96 | 2.96 | 6.98 | 201 | 36.9 | 56.6 | 38.5 | 34.1 | 157 | 1136 |
| residential cooking (g/kg oil) | | - | - | - | - | - | - | - | - | - | - |
| canteen cooking (g/meal) | | 0.00423 | 0.00237 | 0.0142 | 0.0834 | 0.0353 | 0.0389 | 0.0212 | 0.0186 | 0.0711 | 0.359 |
| **uncontrolled EFs** | | | | | | | | | | | |
| commercial cooking (μg/m³) | home-style cuisine | 10.1 | 16.1 | 76.9 | 502 | 119 | 119 | 75.4 | 71.7 | 246 | 699 |
| | Chinese fast food and snacks | 19.1 | 27.3 | 145 | 1312 | 151 | 150 | 197 | 127 | 725 | 4303 |
| | hotpot | 45.5 | 31.8 | 247 | 437 | 238 | 315 | 239 | 273 | 1180 | 4444 |
| | barbecue | 191 | 62.1 | 450 | 432 | 1227 | 649 | 874 | 669 | 4164 | 12840 |
| | Sichuan-Hunan cuisine | 176 | 145 | 1042 | 2669 | 1708 | 2183 | 999 | 1181 | 5129 | 14661 |
| | Cantonese-Fujian cuisine | 31.4 | 29.0 | 154 | 1025 | 289 | 305 | 195 | 167 | 657 | 2975 |
| | Jiangsu-Zhejiang cuisine | 29.1 | 29.3 | 156 | 1125 | 261 | 274 | 200 | 162 | 692 | 3392 |
| | other Chinese cuisines | 43.9 | 38.7 | 206 | 1129 | 423 | 443 | 276 | 255 | 837 | 3440 |
| | non-Chinese cuisines | 24.2 | 21.6 | 48.0 | 1306 | 280 | 456 | 298 | 272 | 1132 | 7371 |
| residential cooking (g/kg oil) | | 0.0989 | 0.295 | 0.452 | 1.12 | 0.874 | 0.946 | 0.528 | 0.674 | 1.80 | 13.5 |
| canteen cooking (g/meal) | | 0.00961 | 0.00535 | 0.0322 | 0.192 | 0.0795 | 0.0877 | 0.0486 | 0.0421 | 0.163 | 0.838 |

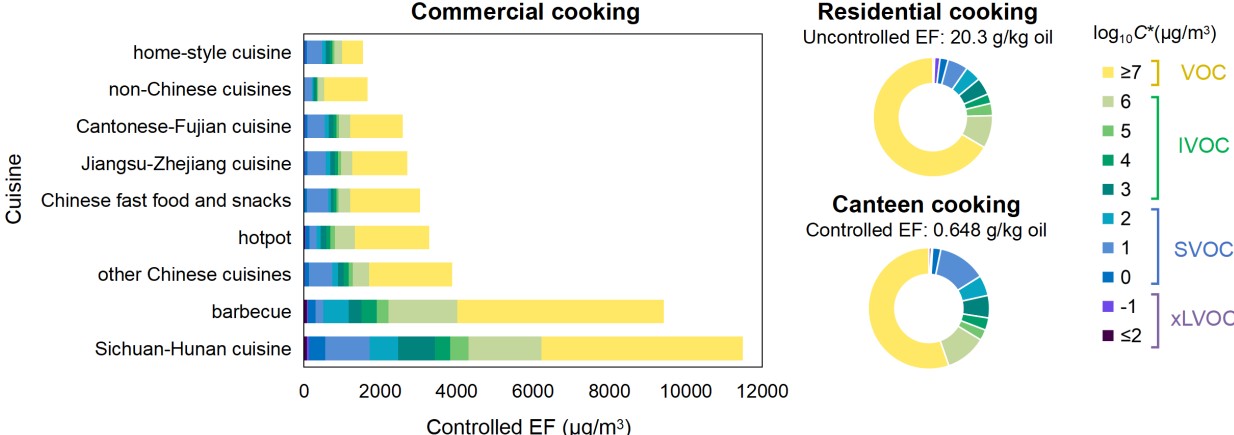

**Figure 1: Full-volatility EFs for different cooking sources.** The colors represent volatility ranges: purple for xLVOCs, blue for SVOCs, green for IVOCs, and yellow for VOCs. Darker colors in the same color group signify lower volatility. The controlled EFs for commercial and canteen cooking are displayed here, as most full-volatility EFs are tested after pollution control facilities and most uncontrolled EFs are inferred from the controlled EFs. The uncontrolled EFs for residential cooking are displayed here, considering the absence of pollution control facilities in home kitchens.

The removal efficiencies of gaseous and particle-phase organics for different cuisines are listed in Table S6. Average removal efficiencies of 57.2% and 55.4% for gaseous and particle-phase organics, respectively. Currently, the national standard and most local standards lack regulations on the removal efficiency for VOCs and particulate matter (PM), let alone for full-volatility organics. Only the local standard of Beijing (Beijing Environmental Protection Bureau, 2018) mentions the removal efficiency requirements of related pollutants, namely non-methane hydrocarbons (NMHC) and PM, as shown in Table S7. In contrast, the current average removal efficiencies of gaseous and particle-phase organics fall short of the efficiencies of NMHC (68.7%) and PM (82.3%) in the Beijing standard (Beijing Environmental Protection Bureau, 2018), possibly due to sub-optimal maintenance and cleaning of fume purification facilities.

Fig. 2 presents the trend of activity data for each emission source from 2015 to 2021, along with the contribution of their respective subsectors averaged over these years. The volume of fume gas produced by commercial restaurants is largely associated with the restaurant number. The total number of restaurants increased from 5.61 million in 2015 to 7.70 million in 2021, reflecting the rapid expansion of China's catering industry. However, there was a dip in the number of restaurants in 2020, potentially attributed to the impact of the COVID-19 pandemic on catering. Furthermore, the cuisine distribution based on point-by-point statistics reveals that Chinese fast food and snacks and home-style cuisine are most common in China during 2015-2021, accounting for 28.3% and 20.7% of total restaurant number, respectively, while non-Chinese cuisines are the least common (3.21%). However, given that most Chinese fast food and snack restaurants are of small scale,



their contribution to the total fume gas volume is much less, accounting for only 11.6%. As the activity data of residential cooking, household edible oil consumption is essentially stable, with fluctuations in certain years. The fluctuations are potentially attributed to less frequent home cooking as the rise of food delivery services, and more frequent home cooking during COVID-19 lockdowns. Additionally, the meals provided by canteens have gradually increased, likely following China's growing population and urbanization.

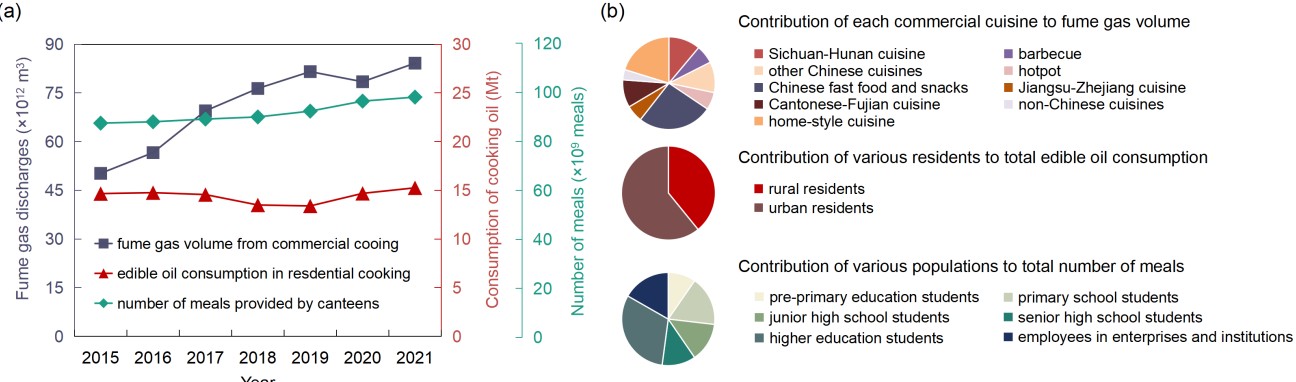


**Figure 2: (a) National activity data for commercial, residential and canteen cooking from 2015-2021, and (b) their sub-sectors' contributions.**

Notably, we obtain point-source precision activity data for commercial cooking, including the geographic location and
cuisine type of each restaurant. This helps accurately determine the spatial distribution of commercial cooking emissions and identify the regional differences. Fig. 3 shows the geographic distribution of all restaurants of various cuisine types across China. The spatial distributions of different cuisines vary greatly. The restaurants serving local specialty cuisines, including Sichuan-Hunan, Fujian-Cantonese, Jiangsu-Zhejiang and other Chinese cuisines (mainly composed of famous Beijing cuisine, Shandong cuisine, Anhui cuisine and Xinjiang cuisine), are clearly concentrated in certain provinces. Restaurants
that serve home-style dishes typically offer common, simple, traditional Chinese meals, similar to those prepared in home kitchens, as opposed to the refined and complex dishes found in larger upscale restaurants. They are not distributed in some provinces because home-style restaurants in these provinces have been classified into local specialty cuisines due to their distinctive local characteristics. Other cuisines are mainly distributed in populous areas throughout the country. Based on Fig. S2, we find that the cuisine distributions are similar in most provinces across China. However, in some provinces, such as
Hunan, Guangdong, Zhejiang, and Beijing, the local specialty cuisines are dominant.

**Figure 3: The geographical location of all restaurants in China, categorized by cuisines.**

Table 3 illustrates the PFIPs of 31 provinces in China from 2015 to 2021, where a darker cell color in the table indicates a lower PFIP value. An overall improvement in pollution control has been observed nationwide in recent years as the importance of cooking emissions has been increasingly recognized. The national overall PFIP increased from 54.4% in 2015 to 73.9% in 2021. Besides, the PFIP tends to increase with restaurant scale, probably because larger restaurants face greater regulatory pressure or have more funding for fume purification. The PFIP exhibits considerable variation across different

provinces, due to different levels of focus on pollution control in the catering industry. Provinces such as Beijing, Liaoning, Shanghai, and Hainan were the first to achieve PFIPs of 100%, as they have long emphasized the control of pollution




emissions from cooking sources and have issued explicit regulations requiring all restaurants to install purification facilities. In contrast, the PFIP in most other provinces remains relatively low, as major efforts to strengthen cooking source pollution control in these regions were initiated mainly between 2015 and 2018.

**Table 3: Purification facility installation proportion for restaurants of different scales in each province.**

| scale / year province | large | | | | | | | medium | | | | | | | small | | | | | | |
|---|---|---|---|---|---|---|---|---|---|---|---|---|---|---|---|---|---|---|---|---|---|
| | 2015 | 2016 | 2017 | 2018 | 2019 | 2020 | 2021 | 2015 | 2016 | 2017 | 2018 | 2019 | 2020 | 2021 | 2015 | 2016 | 2017 | 2018 | 2019 | 2020 | 2021 |
| Beijing | 82.8% | 82.8% | 82.8% | 88.5% | 94.3% | 100% | 100% | 72.0% | 72.0% | 72.0% | 81.3% | 90.7% | 100% | 100% | 60.0% | 60.0% | 60.0% | 73.3% | 86.7% | 100% | 100% |
| Tianjin | 82.8% | 88.5% | 94.3% | 100% | 100% | 100% | 100% | 72.8% | 81.9% | 90.9% | 100% | 100% | 100% | 100% | 59.9% | 73.2% | 86.6% | 100% | 100% | 100% | 100% |
| Hebei | 82.8% | 82.8% | 88.5% | 94.3% | 100% | 100% | 100% | 72.8% | 72.8% | 81.9% | 90.9% | 100% | 100% | 100% | 59.9% | 59.9% | 73.2% | 86.6% | 100% | 100% | 100% |
| Shanxi | 82.8% | 82.8% | 82.8% | 82.8% | 82.8% | 82.8% | 82.8% | 72.8% | 72.8% | 72.8% | 72.8% | 72.8% | 72.8% | 72.8% | 59.9% | 59.9% | 59.9% | 59.9% | 59.9% | 59.9% | 59.9% |
| Inner Mongolia | 64.0% | 70.3% | 76.5% | 82.8% | 82.8% | 82.8% | 82.8% | 59.0% | 63.6% | 68.2% | 72.8% | 72.8% | 72.8% | 72.8% | 41.0% | 47.3% | 53.6% | 59.9% | 59.9% | 59.9% | 59.9% |
| Liaoning | 82.8% | 88.5% | 94.3% | 100% | 100% | 100% | 100% | 72.8% | 81.9% | 90.9% | 100% | 100% | 100% | 100% | 59.9% | 73.2% | 86.6% | 100% | 100% | 100% | 100% |
| Jilin | 64.0% | 64.0% | 64.0% | 70.3% | 76.5% | 82.8% | 82.8% | 59.0% | 59.0% | 59.0% | 63.6% | 68.2% | 72.8% | 72.8% | 41.0% | 41.0% | 41.0% | 47.3% | 53.6% | 59.9% | 59.9% |
| Heilongjiang | 64.0% | 70.3% | 76.5% | 82.8% | 82.8% | 82.8% | 82.8% | 59.0% | 63.6% | 68.2% | 72.8% | 72.8% | 72.8% | 72.8% | 41.0% | 47.3% | 53.6% | 59.9% | 59.9% | 59.9% | 59.9% |
| Shanghai | 94.3% | 100% | 100% | 100% | 100% | 100% | 100% | 90.9% | 100% | 100% | 100% | 100% | 100% | 100% | 86.6% | 100% | 100% | 100% | 100% | 100% | 100% |
| Jiangsu | 64.0% | 70.3% | 76.5% | 82.8% | 82.8% | 82.8% | 88.5% | 59.0% | 63.6% | 68.2% | 72.8% | 72.8% | 72.8% | 81.9% | 41.0% | 47.3% | 53.6% | 59.9% | 59.9% | 59.9% | 73.2% |
| Zhejiang | 64.0% | 70.3% | 76.5% | 82.8% | 82.8% | 82.8% | 82.8% | 59.0% | 63.6% | 68.2% | 72.8% | 72.8% | 72.8% | 72.8% | 41.0% | 47.3% | 53.6% | 59.9% | 59.9% | 59.9% | 59.9% |
| Anhui | 64.0% | 70.3% | 76.5% | 82.8% | 82.8% | 82.8% | 82.8% | 59.0% | 63.6% | 68.2% | 72.8% | 72.8% | 72.8% | 72.8% | 41.0% | 47.3% | 53.6% | 59.9% | 59.9% | 59.9% | 59.9% |
| Fujian | 64.0% | 70.3% | 76.5% | 82.8% | 82.8% | 82.8% | 82.8% | 59.0% | 63.6% | 68.2% | 72.8% | 72.8% | 72.8% | 72.8% | 41.0% | 47.3% | 53.6% | 59.9% | 59.9% | 59.9% | 59.9% |
| Jiangxi | 64.0% | 64.0% | 70.3% | 76.5% | 82.8% | 82.8% | 82.8% | 59.0% | 59.0% | 63.6% | 68.2% | 72.8% | 72.8% | 72.8% | 41.0% | 41.0% | 47.3% | 53.6% | 59.9% | 59.9% | 59.9% |
| Shandong | 64.0% | 64.0% | 70.3% | 76.5% | 82.8% | 82.8% | 82.8% | 59.0% | 59.0% | 63.6% | 68.2% | 72.8% | 72.8% | 72.8% | 41.0% | 41.0% | 47.3% | 53.6% | 59.9% | 59.9% | 59.9% |
| Henan | 82.8% | 82.8% | 82.8% | 88.5% | 94.3% | 100% | 100% | 72.8% | 72.8% | 72.8% | 81.9% | 90.9% | 100% | 100% | 59.9% | 59.9% | 59.9% | 73.2% | 86.6% | 100% | 100% |
| Hubei | 64.0% | 64.0% | 70.3% | 76.5% | 82.8% | 82.8% | 82.8% | 59.0% | 59.0% | 63.6% | 68.2% | 72.8% | 72.8% | 72.8% | 41.0% | 41.0% | 47.3% | 53.6% | 59.9% | 59.9% | 59.9% |
| Hunan | 64.0% | 64.0% | 70.3% | 76.5% | 82.8% | 82.8% | 82.8% | 59.0% | 59.0% | 63.6% | 68.2% | 72.8% | 72.8% | 72.8% | 41.0% | 41.0% | 47.3% | 53.6% | 59.9% | 59.9% | 59.9% |
| Guangdong | 82.8% | 82.8% | 82.8% | 82.8% | 82.8% | 82.8% | 82.8% | 72.8% | 72.8% | 72.8% | 72.8% | 72.8% | 72.8% | 72.8% | 59.9% | 59.9% | 59.9% | 59.9% | 59.9% | 59.9% | 59.9% |
| Guangxi | 64.0% | 64.0% | 70.3% | 76.5% | 82.8% | 82.8% | 82.8% | 59.0% | 59.0% | 63.6% | 68.2% | 72.8% | 72.8% | 72.8% | 41.0% | 41.0% | 47.3% | 53.6% | 59.9% | 59.9% | 59.9% |
| Hainan | 100% | 100% | 100% | 100% | 100% | 100% | 100% | 100% | 100% | 100% | 100% | 100% | 100% | 100% | 100% | 100% | 100% | 100% | 100% | 100% | 100% |
| Chongqing | 82.8% | 82.8% | 82.8% | 88.5% | 94.3% | 100% | 100% | 72.8% | 72.8% | 72.8% | 81.9% | 90.9% | 100% | 100% | 59.9% | 59.9% | 59.9% | 73.2% | 86.6% | 100% | 100% |
| Sichuan | 64.0% | 70.3% | 76.5% | 82.8% | 82.8% | 82.8% | 82.8% | 59.0% | 63.6% | 68.2% | 72.8% | 72.8% | 72.8% | 72.8% | 41.0% | 47.3% | 53.6% | 59.9% | 59.9% | 59.9% | 59.9% |
| Guizhou | 64.0% | 70.3% | 76.5% | 82.8% | 82.8% | 82.8% | 82.8% | 59.0% | 63.6% | 68.2% | 72.8% | 72.8% | 72.8% | 72.8% | 41.0% | 47.3% | 53.6% | 59.9% | 59.9% | 59.9% | 59.9% |
| Yunnan | 64.0% | 64.0% | 64.0% | 64.0% | 70.3% | 76.5% | 82.8% | 59.0% | 59.0% | 59.0% | 59.0% | 63.6% | 68.2% | 72.8% | 41.0% | 41.0% | 41.0% | 41.0% | 47.3% | 53.6% | 59.9% |
| Xizang | 64.0% | 64.0% | 64.0% | 64.0% | 70.3% | 76.5% | 82.8% | 59.0% | 59.0% | 59.0% | 59.0% | 63.3% | 67.7% | 72.0% | 41.0% | 41.0% | 41.0% | 41.0% | 47.3% | 53.6% | 59.9% |
| Shaanxi | 64.0% | 64.0% | 64.0% | 70.3% | 76.5% | 82.8% | 82.8% | 59.0% | 59.0% | 59.0% | 63.3% | 67.7% | 72.0% | 72.0% | 41.0% | 41.0% | 41.0% | 47.3% | 53.6% | 59.9% | 59.9% |
| Gansu | 64.0% | 64.0% | 64.0% | 64.0% | 70.3% | 76.5% | 82.8% | 59.0% | 59.0% | 59.0% | 59.0% | 63.3% | 67.7% | 72.0% | 41.0% | 41.0% | 41.0% | 41.0% | 47.3% | 53.6% | 59.9% |
| Qinghai | 64.0% | 64.0% | 64.0% | 64.0% | 70.3% | 76.5% | 82.8% | 59.0% | 59.0% | 59.0% | 59.0% | 63.3% | 67.7% | 72.0% | 41.0% | 41.0% | 41.0% | 41.0% | 47.3% | 53.6% | 59.9% |
| Ningxia | 64.0% | 64.0% | 64.0% | 64.0% | 64.0% | 70.3% | 76.5% | 59.0% | 59.0% | 59.0% | 59.0% | 59.0% | 63.3% | 67.7% | 41.0% | 41.0% | 41.0% | 41.0% | 41.0% | 47.3% | 53.6% |
| Xinjiang | 64.0% | 64.0% | 64.0% | 64.0% | 70.3% | 76.5% | 82.8% | 59.0% | 59.0% | 59.0% | 59.0% | 63.3% | 67.7% | 72.0% | 41.0% | 41.0% | 41.0% | 41.0% | 47.3% | 53.6% | 59.9% |

**3.2 Full-volatility organic emissions from cooking in China**

Fig. 4 shows the national emission inventory of full-volatility organic emissions from cooking in 2021 and the uncertainty ranges of emissions. The xLVOC, SVOC, IVOC, and VOC emissions in China in 2021 are 13.1 (7.36-21.0, 95% confidence level) kt/yr, 176 (95.8-290) kt/yr, 241 (135-374) kt/yr, and 561 (317-891) kt/yr, respectively. The majority of these emissions are VOCs (56.4%), followed by IVOCs (24.6%) and SVOCs (17.7%), with xLVOCs comprising only 1.32%.



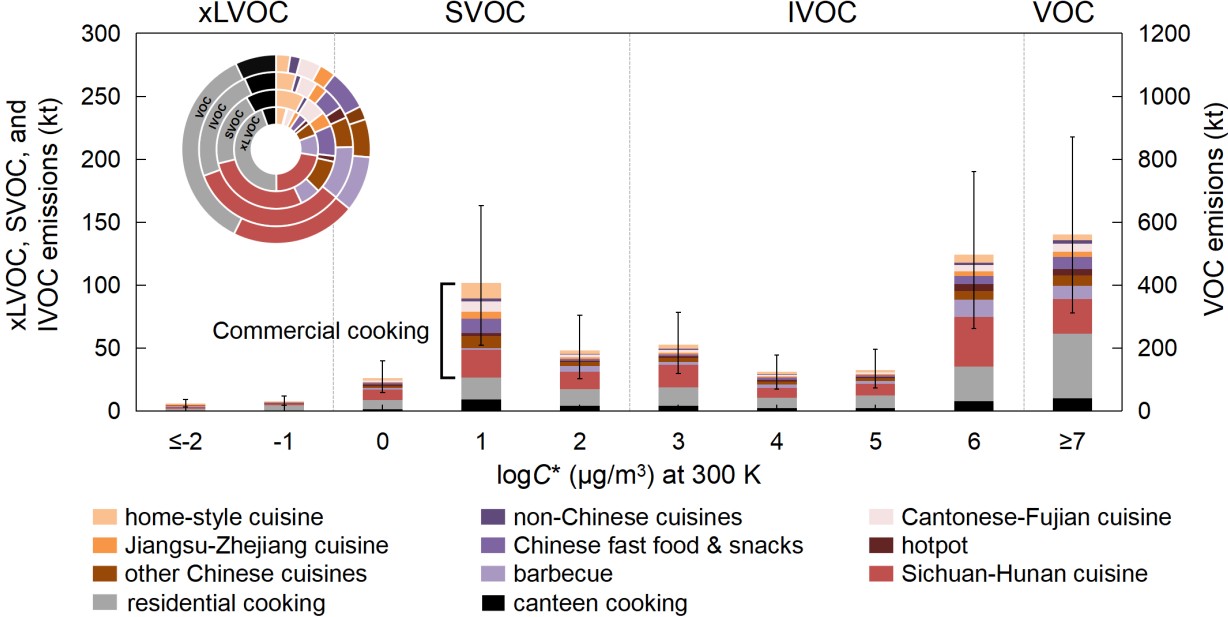

**Figure 4: Full-volatility organic emissions from different cooking sources in China in 2021.** The gray and black bars
represent residential and canteen cooking emissions in each volatility bin, respectively. Other colors represent the emissions
from different commercial cuisines. Error bars indicate the uncertainty range of emissions under a 95% confidence level. The
upper left corner displays the proportionate contributions to emissions across the four volatility ranges from various types of
cooking sources.

Among all cooking sources, commercial cooking has the most prominent emissions, contributing 46.7%, 68.5%, 66.2%, and
54.5%, respectively, to the xLVOC, SVOC, IVOC, and VOC emissions on average from 2015 to 2021. Sichuan-Hunan
cuisine contributes the most to emissions among all commercial cuisines, accounting for 19.3%-30.6% of cooking emissions
of all cooking sources in the four volatility ranges, despite Sichuan-Hunan cuisine not being the most common (making up
only 11.1% of all restaurants). In contrast, the most common Chinese fast food and snacks and home-style cuisine contribute
less to the emissions (≤9.25%) in the four volatility ranges. This further emphasizes the significant influence of variations in
EFs across different cuisines. Additionally, residential cooking is also a notable source, contributing 47.1%, 22.2%, 25.9%,
and 37.5% to xLVOC, SVOC, IVOC, and VOC emissions, respectively, whereas canteens make minor contributions to full-
volatility organics (<10%).

The uncertainty ranges (95% confidence interval) of the national cooking emissions are estimated at [-47.5%, 60.2%] for
commercial cooking, [-63.0%, -124%] for residential cooking, [-91.0%, -213%] for canteen cooking, and [-45.2%, +53.5%]



for total cooking emissions. Total emissions' uncertainty is less due to offset effects across sectors. The relatively large uncertainty in canteen emissions arises from activity level estimates and EF unit conversions. As canteen emissions are small, their uncertainty has little impact on that of total emissions. Residential emission uncertainty also largely originates from EF unit conversions, while commercial cooking's smaller uncertainty is due to its more EF tests and better statistics.

Furthermore, we evaluate the importance of cooking organic emissions by supplementing our emission inventory in 2017 into the full-volatility emissions inventory in 2017 developed by Chang et al. (2022), which lacks cooking sources. The results indicate that the cooking source contributes 1.03%, 12.7%, 5.53%, and 1.83% to total xLVOC, SVOC, IVOC, and VOC emissions, respectively. This reveals the significance of I/SVOC emissions from the cooking source, suggesting that accounting for the previously missing cooking source may be crucial for accurately identifying the source of SOA. In fact,

cooking activities are often concentrated in densely populated urban areas. Table 4 lists the contributions of cooking emissions to the total emissions in the four volatility ranges in the five most densely populated cities in China. In these regions, the importance of organic emissions from cooking, particularly I/SVOC emissions, is much higher than the national average. The contributions to SVOC emissions from cooking sources are all above 30%, reaching up to 61.7% at maximum. The contribution of cooking sources to IVOCs is also significant (9.34%-21.7%). Furthermore, the close affinity of cooking

activities with the human living environment renders its organic emissions a high health risk. Therefore, obtaining accurate cooking emissions, including their spatial distribution, is necessary for studies on the causes and health impacts of air pollution in the human living environment.

**Table 4: The contributions of cooking emissions to the total emissions in the four volatility ranges in the five most densely populated cities in China.**

| city, province | the contributions of cooking organic emissions to the total emissions | | | |
| --- | --- | --- | --- | --- |
| | xLVOC | SVOC | IVOC | VOC |
| Shenzhen, Guangdong | 5.03% | 44.7% | 12.1% | 1.84% |
| Dongguan, Guangdong | 9.53% | 61.7% | 21.7% | 3.55% |
| Shanghai, Shanghai | 8.22% | 43.4% | 10.5% | 1.03% |
| Xiamen, Fujian | 2.23% | 31.3% | 9.34% | 2.03% |
| Guangzhou, Guangdong | 5.78% | 48.6% | 13.4% | 2.14% |


### 3.3 Spatial distributions of emissions

The comprehensive and cuisine-specific activity data in our emission estimates, coupled with the provincial policy-driven PFIPs, allow us to discern regional emission disparities accurately. Fig. 5 displays the provincial total and per capita emissions from cooking sources in China in 2021. The provincial total emissions are closely associated with population. The

provinces with the highest populations—Guangdong, Shandong, Henan, and Jiangsu—are at the forefront of emissions, whereas those with the smallest populations—Tibet, Qinghai, Ningxia, and Hainan—are at the bottom. Surprisingly, per

capita emissions show a three-fold difference among provinces, likely attributed to different dietary preferences. For example, people in Sichuan and Hunan prefer spicy and oil-rich food, increasing average commercial cooking EFs and household edible oil consumption in these regions. Therefore, the per capita emissions in Hunan and Sichuan (1.35 and 1.19

kg/person, respectively) are significantly higher than the national average (0.701 kg/person). Moreover, the importance of emission sources varies by province, but the overall picture across all provinces is that commercial cooking emissions are generally the most prominent, followed by domestic cooking, with minimal emissions from canteen cooking.

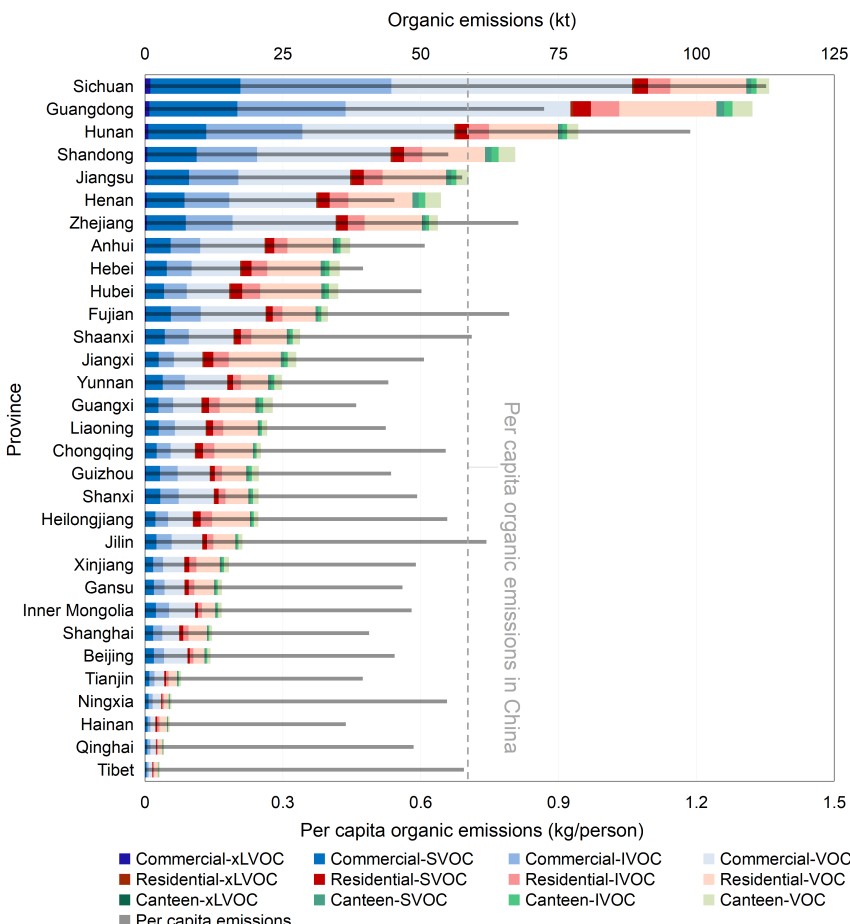

**Figure 5: Provincial total emissions and per capita emissions across China in 2021.** The thick bars represent the total

emissions for each province. The blue, red, and green bars represent organic emissions from commercial cooking, home cooking, and canteen cooking sources, respectively. Within the same color group, four different shades represent different volatility ranges of organic matter, namely xLVOC, SVOC, IVOC, and VOC, with darker colors indicating lower volatility. The thin gray bars represent per capita organic emissions in each province. The gray dashed line represents the national per capita organic emissions.




To identify high-emission areas and hotspots, we have further allocated cooking emissions, including commercial cooking emissions with point-source precision and residential and canteen cooking emissions, into grids at 27 km × 27 km resolution (Fig. 6). As previously analyzed, high population density and specific dietary preferences are two important features of high-emission areas. Representative areas of high population density include North China Plain (NCP), Yangtze River Delta (YRD), and Pearl River Delta (PRD), indicated by red circles. Besides, capital cities in central and eastern Chinese provinces also emerge as emission hotspots due to high population densities. The large population in these areas fosters a flourishing commercial catering industry and substantial residential cooking, thereby producing significant emissions. Besides, high emissions are present in Sichuan (SC) and Hunan (HN), highlighted by green circles. These regions not only have significant populations and prosperous catering industries that hold appeal for people nationwide, but more importantly, the spice-rich and oily characteristic of the local food amplifies the emissions.

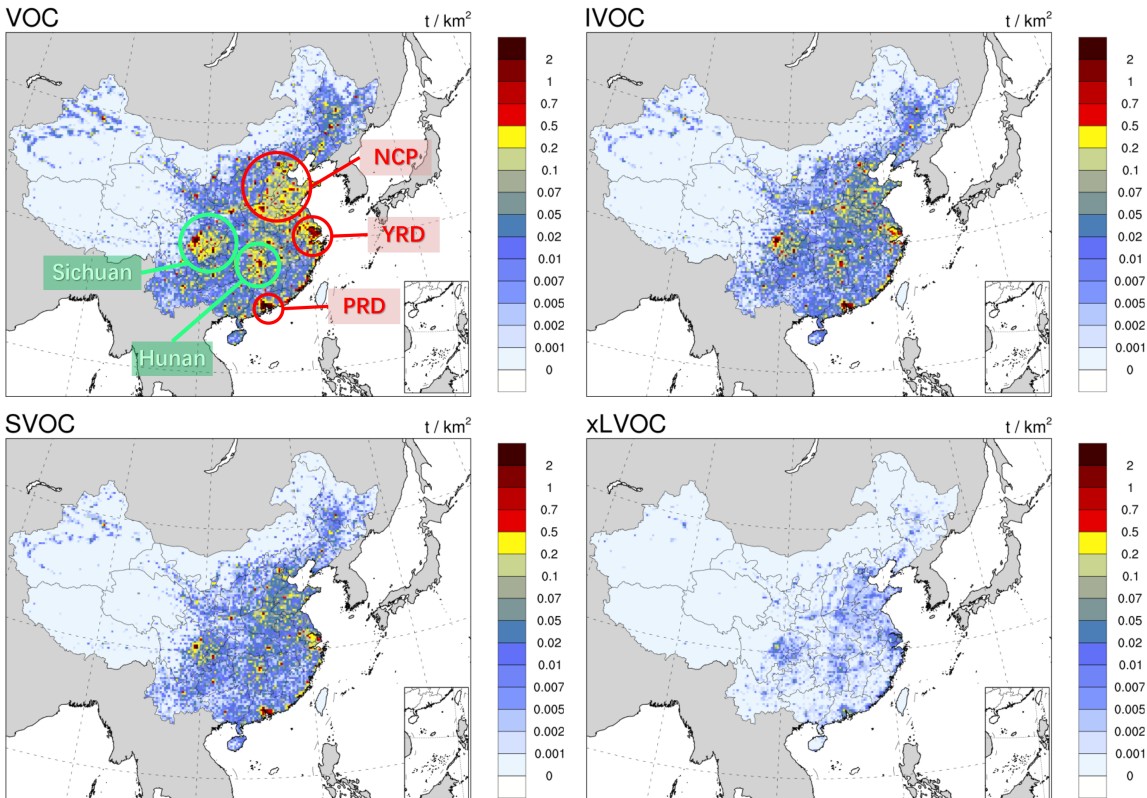

**Figure 6: Nationwide gridded cooking emissions in the four volatility ranges in 2021, with high-emission areas circled.**



### 3.4 Historical trends and drivers of emissions

Fig. 7a displays cooking emissions from various sources during 2015-2021. Overall, the total cooking organic emissions

have slowly increased from 791 kt in 2015 to 990 kt in 2021. This upward trend is mainly due to the overall growth in commercial cooking emissions, which increased from 414 kt in 2015 to 609 kt in 2021, while emissions from residential and canteen cooking only fluctuated slightly. Besides, the percentages of xLVOC, SVOC, IVOC, and VOC during these years are generally stable, with an average of 1.32%, 17.51%, 24.2%, and 56.9%.

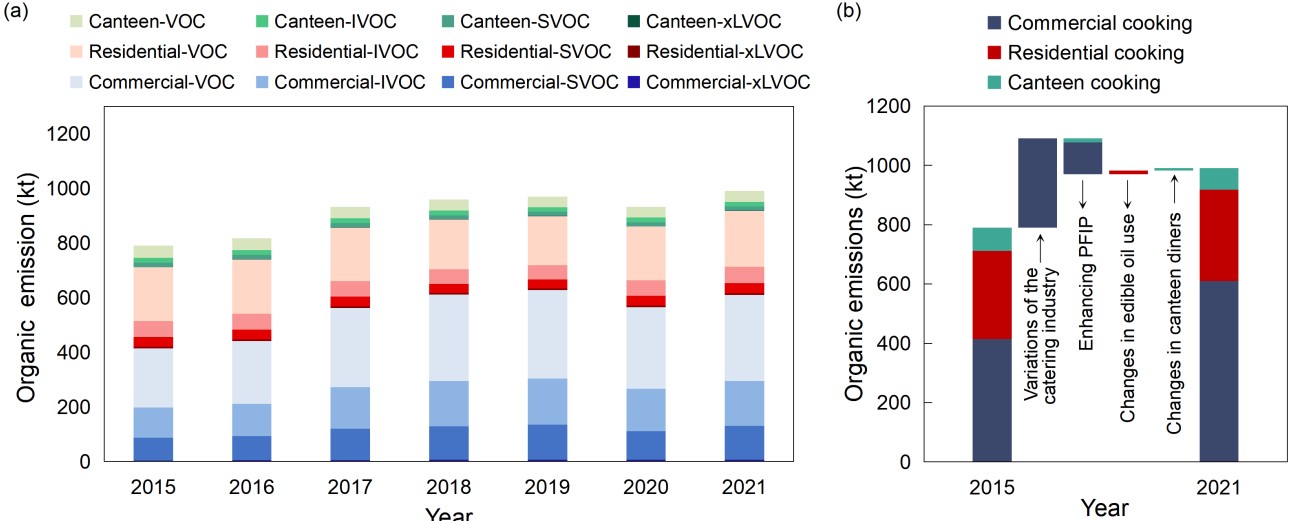

**Figure 7: (a) organic emissions in the four volatility ranges from various cooking sources in 2015-2021, and (b) contributions of different drivers to the changes of organic emissions from 2015 to 2021.**

Fig. 7b illustrates the contribution of four influencing factors mentioned in Section 2.5 to the changes in organic emissions from 2015 to 2021 (see Fig. S3 for the annual contributions of different factors). The development of the catering industry

drives the overall increase in cooking emissions, leading to an average annual emission growth rate of 6.36% from 2015 to 2021. However, the case in 2020 is an exception, when the variation of the catering industry regressed and the cooking emissions were reduced by 4.95% due to the COVID-19 lockdown measure. Besides, the yearly increase in the installation proportion of purification facilities could mitigate emissions, and it resulted in an average annual emission reduction rate of 2.25% from 2015 to 2021. However, its effect was limited in comparison to the rapid development of the catering industry

due to inadequate regulations on cooking emissions nationwide. Furthermore, the overall impact of household edible oil consumption is minor in 2015-2021, but in years with significant changes in dietary habits, it could cause notable shifts in emissions. For instance, in 2020, the change in household edible oil consumption caused a 2.63% increase in total emissions





due to an increase in household cooking under the COVID-19 lockdown measures. Additionally, the meals provided by canteens have gradually increased in recent years, but its impact on overall emission changes is minimal, due to the small
scale of canteen emissions.

**3.5 Comparison with other related emission inventories**

Table 5 compares cooking emissions at different spatial scales in our study and previous studies (Lin et al., 2022; Liang et al., 2022; Qi et al., 2020; Jin et al., 2021; Wang et al., 2018a; Yuan et al., 2023; Cheng et al., 2022). Previous inventories mainly considered pollutants such as VOCs, PM$_{2.5}$, and OC, all of which we convert to VOCs or POA for comparison. In contrast,
our inventory manages to cover full-volatility organics, comprehensive cooking sources, a wide time range and various regional scales, which were difficult to achieve in previous inventories.

**Table 5: Comparison of cooking emissions in this study with those in previous studies.** Bolded words represent this study.

| region and year | inventory studies | VOC emissions (kt) | | | POA emissions (kt) | | |
|---|---|---|---|---|---|---|---|
| | | commercial cooking | residential cooking | canteen cooking | commercial cooking | residential cooking | canteen cooking |
| **China, 2015[a]** | **this study** | **216** | **197** | **43.8** | **132** | **96.8** | **33.0** |
| China, 2012 | Wang et al., 2018a | 66.0 | | | | | |
| **China, 2017** | **this study** | **290** | **196** | **42.1** | **180** | **96.1** | **31.8** |
| China, 2017 | Jin et al., 2021 | 34.0 | | | | | |
| **China, 2018** | **this study** | **317** | **183** | **40.3** | **195** | **89.5** | **30.4** |
| China, 2018 | Cheng et al., 2022 (POA=1.8OC (Huang, 2023)) | | | | 2.31 | 2.22 | |
| **China, 2019** | **this study** | **325** | **180** | **39.6** | **201** | **88.1** | **29.9** |
| China, 2019 | Liang et al., 2022 | 93.8 | 94.8 | 45.1 | | | |
| **Beijing, 2018** | **this study** | **6.15** | **2.13** | **0.699** | **4.01** | **1.04** | **0.528** |
| Beijing, 2019 | Lin et al., 2022 (POA=81.5%PM$_{2.5}$) | 3.60 | | | 1.43 | | |
| Beijing, 2018 | Qi et al., 2020 (POA=81.5%PM$_{2.5}$) | | | | | 0.45 | |
| **Shanghai, 2015[a]** | **this study** | **2.40** | **3.22** | **0.499** | | | |
| Shanghai, 2012 | Wang et al., 2018a | 4.69 | 0.61 | 1.05 | | | |
| **Shunde, 2018** | **this study** | **0.730** | **0.560** | **0.0812** | **0.660** | **0.270** | **0.0641** |
| Shunde, 2018 | Yuan et al., 2023 (POA=81.5%PM$_{2.5}$) | 1.26 | | | 1.18 | | |

[a] Due to data limitations, our inventory only covers emissions up to 2015, so the earliest available results from 2015 are used
to compare with the 2012 results reported by Wang et al. (2018a).



At the national scale, our estimate of cooking emissions is significantly higher than previous calculations (Lin et al., 2022; Wang et al., 2018a; Yuan et al., 2023; Cheng et al., 2022). This significant discrepancy is probably due to potential omissions in activity data and biases introduced by inaccurate EFs and PFIPs in previous studies. Specifically, Cheng et al.'s (2022) estimations based on meat consumption were 98.4% lower than our estimate. This is because cooking emissions are not only determined by meat consumption, but also involve the use of vegetables, cooking oils, and condiments, so the underrepresented activity data could introduce large errors. Commercial cooking emissions estimated by Jin et al. (2021) are also 88.3% lower than our estimates, possibly because they used EFs that were not based on measurements and applied uniform EFs across China without distinguishing between cuisines, introducing a significant uncertainty. In comparison, the estimations from Wang et al. (2018a) and Liang et al. (2022) differ less significantly from our estimates, with their results being 69.4% and 57.1% lower, respectively, possibly because they used cuisine-specific EFs derived from measurements, thereby improving the accuracy of the EFs. However, applying controlled EFs in all restaurants, including those without pollution control facilities, probably led to an underestimation of their emissions.

At the city and district scales, the previous inventories (Lin et al., 2022; Qi et al., 2020; Wang et al., 2018a; Yuan et al., 2023) were calculated in a more refined way. Therefore, our results are closer to previous estimates. However, differences persist due to uncertainties in our calculations and those of previous studies. Notably, our estimated emissions agree remarkably well with the emission inventory based on the online monitoring system in Shunde, as both inventories use refined point-source data as activity data. Overall, our study achieves broader coverage across multiple dimensions and significantly rectifies the previous underestimations in national inventories.

## 4 Data availability

The full-volatility organic emissions dataset is available at https://doi.org/10.6084/m9.figshare.23537673 (Li et al., 2023). It includes multi-year provincial full-volatility emissions from residential cooking, canteen cooking, and cuisine-specific commercial cooking. For commercial cooking emissions, we also provide full-volatility emissions with point-source accuracy. Besides, the dataset also provides gridded emissions in China for xLVOCs, SVOCs, IVOCs and VOCs from the three cooking sources at a resolution of 27 km × 27 km. Emission factors, PFIP, and other calculated parameters used for emission estimates are listed in the main text and supplementary materials. The catering-related POI data is obtained from the Amap map service (https://lbs.amap.com/, Amap, 2022). In addition, the required statistical data, including provincial populations, the number of students in different stages of education, the number of employees in state-owned and collective enterprises and institutions, urban and rural populations, per capita consumption of household cooking oil in urban and rural residents, city population and city area, are obtained from the China Statistical Yearbook and China Labour Statistical Yearbook at https://data.stats.gov.cn/ (National Bureau of Statistics, 2022a,b).





## 5 Conclusions and implication

Existing cooking inventories rarely covered full-volatility organics and failed to achieve accurate emission estimates with high resolution at a national scale, preventing an accurate understanding of the characteristics and health impacts of cooking
emissions. Our study fills this gap by developing a high-resolution national inventory of full-volatility organic emissions from cooking in China. The state-of-the-art inventory updates the understanding of characteristics, sources, and regional variations of cooking emissions across China. The emissions of xLVOC, SVOC, IVOC and VOC from cooking in China in 2021 were 13.1 kt/yr, 176 kt/yr, 241 kt/yr, and 561 kt/yr, respectively. It reveals that the IVOCs and SVOCs emitted from cooking sources are of great importance, especially in densely populated cities where they account for 9-21% and 31-62% of
the total IVOC and SVOC emissions from all sources, thereby potentially greatly impacting SOA formation and human health. Besides, our inventory comprehensively includes emissions from home kitchens, canteens, and restaurants with various cuisines, and also corrects significant underestimations in previous emission calculations for these sources due to potential omission of activity data as well as the oversimplified EFs and PFIPs, which aids in accurate identification and effective control of emission sources. We find that commercial and residential cooking are two important sources,
contributing over 90% of total organic emissions from cooking. Moreover, we find that local dietary habits significantly influence cooking emissions. For example, in areas where spicy and oily foods are preferred, the per capita organic emissions from cooking (1.19-1.35 kg/person) are much higher than the average (0.701 kg/person). Such regional features would be obscured when using a national uniform EF. Overall, our data set provides meaningful information for precise regulation of organic cooking emissions (including gaseous and particle-phase organics) in China, and also provides the
prerequisite for the accurate modeling of SOA formation and evolution.

Based on the multi-year national cooking emission inventory and sensitivity analyses, we discover that despite annual increases in PFIPs, they cannot offset the emission increases caused by the rapid growth of the catering industry. Given the significant health risks potentially posed by cooking emissions, future efforts to reduce cooking emissions need to be strengthened through multiple pathways. Considering that the overall PFIP for restaurants nationwide in 2021 is only about
73.9%, the continued promotion of purification facilities remains a critical emission reduction strategy. Moreover, it may be more important to ensure that the installed purification facilities meet the removal efficiency requirements. According to our estimates, total cooking emissions could be reduced by about 30% if the current removal efficiencies for gaseous and particle-phase organics met the standards for NMHC and PM, respectively (Beijing Environmental Protection Bureau, 2018). Furthermore, residential cooking is also an important emission source, but currently, it lacks dedicated purification facilities,
so it may have a great emission reduction potential. Consideration could be given to equipping residential chimneys with uniform flue gas purifiers or developing miniature fume purifiers that could be installed at the back end of home kitchen range hoods. Our methodology and integrated parameters allow the emission inventories to be extended to different locations and times, and can be used to predict the effect of emission reductions in future control scenarios to evaluate the effectiveness of control strategies.

We also acknowledge some limitations of our study. Due to the potential inadequacy of earlier digital map construction, we can only guarantee reliable information on commercial restaurants from 2015 onwards. While it's challenging to retrospectively track high-resolution emissions using our methodology, pre-2015 emissions could be estimated using the data from 2015 to 2021 and previous statistical data. Additionally, due to limited full-volatility tests and basic data for the cooking activities, we have made some estimates and supplements, which may introduce some uncertainties. Nonetheless, as

the first national inventory of full-volatility organic cooking emissions, our dataset provides many novel and meaningful insights within an acceptable uncertainty range. In the future, further measurements of full-volatility EFs and surveys of cooking habits and fume purification facility installations may help reduce these uncertainties.

**Author contributions**

Zeqi Li, Shuxiao Wang, Shengyue Li, Bin Zhao designed the study. Zeqi Li developed the emission inventory; Shuxiao Wang, Shengyue Li, Xiaochun Wang, Guanghan Huang, Xing Chang, Lyuyin Huang, and Bin Zhao helped to improve the emission inventory. Zeqi Li and Xiaochun Wang did the spatial distribution of the emission inventory. Zeqi Li wrote the original draft; all the coauthors revised the manuscript.

**Competing interests**

The authors declare that they have no conflict of interest.

**Disclaimer**

Publisher's note: Copernicus Publications remains neutral with regard to jurisdictional claims in published maps and

institutional affiliations.

**Acknowledgements**

This work was supported by National Natural Science Foundation of China (92044302, 22188102), Samsung Advanced Institute of Technology, and Tsinghua-Toyota Joint Research Institute Inter-disciplinary Program.


**Financial support**

This study was jointly funded by National Natural Science Foundation of China (92044302, 22188102), Samsung Advanced Institute of Technology, and Tsinghua-Toyota Joint Research Institute Inter-disciplinary Program.



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
