# Peer review of "High-resolution emission inventory of full-volatility organic compounds from cooking in China during 2015-2021"

_Earth System Science Data, 2023_

## Author Comment (AC1)

**September 26, 2023**

**Dear editor and the anonymous reviewers,**

Thanks a lot for your work and time on our manuscript.

The paper entitled ***"High-resolution emission inventory of full-volatility organic compounds from cooking in China during 2015-2021"*** (**Manuscript ID: essd-2023-278**) by Zeqi Li, et al., has been revised carefully according to the correction requests and review reports.

The authors have addressed all the reviews' comments point-by-point as below. All the corrections and responses have been incorporated into the revised manuscript and supplement (marked with **BLUE COLORED FONTS**).

If further responses and corrections should be made, please don't hesitate to let us know.

Sincerely,
Bin Zhao, Assistant Professor
School of Environment
Tsinghua University, Beijing 100084, China
Email: bzhao@mail.tsinghua.edu.cn

**Reply on Referee #1:**

Dear reviewer,

Thank you very much for your recognition and the valuable suggestions! We have addressed the comments point-by-point as below.

All the corrections have been incorporated into the revised manuscript (manuscript_R1) and the revised supplement (supplement_R1). The point-to-point responses are listed as followed. If further responses and corrections should be made, please don't hesitate to let us know.

**Comment 1:**

The authors should give more information about Chinese cooking to make the manuscript more understandable and appealing to global readers. The unique characteristics of Chinese cooking, especially the differences from the cooking of other countries, should be pointed out in the introduction section to show why they are worth considering. The complexity of the cuisines and the variety of compounds mentioned so far (Line 57-59) do not seem to adequately characterize the uniqueness of Chinese cooking. In the methodology section, when discussing the classification of cuisines (Line 167-170), the reasons for such a categorization also need to be detailed, and a brief description of the characteristics of each cuisines category should be included in the main text to make it comprehensible to readers who are unfamiliar with the characteristics of Chinese cuisine.

**Response 1:**

Thank you for the insightful comments and suggestions to help enhance the international appeal of our manuscript. Based on your suggestions, we have supplemented the relevant information about Chinese cooking in the corresponding section.

(1) We have expanded the introduction section to highlight the unique characteristics of Chinese cuisine, emphasizing its differences from cuisines in other countries. We have added the following to the introduction (**lines 54-61**) of the manuscript_R1: **"Furthermore, Chinese cooking stands distinct from those of other countries due to its cuisine diversity and unique cooking styles (Zhao and Zhao, 2018). With the vast regional variation, various popular Chinese cuisines such as Sichuan cuisine and Cantonese cuisine have flourished, each having distinct cooking methods and ingredients (Lin et al., 2022; Liang et al., 2022). This diversity results in significant variations in emission characteristics. Additionally, compared to western cooking methods, the common practice of using oil at high**

**temperatures in Chinese cooking, especially the widespread technique of high-temperature stir-frying (Chen et al., 2018; Liang et al., 2022), might result in a more complex emission of organic compounds (Zhao et al., 2018). Therefore, the unique characteristics and significance of Chinese cooking warrant special attention."** We believe this addition will help readers better understand why Chinese cooking is worthy of consideration.

(2) In the methodology section, we have elaborated on the reasons for categorizing the cuisines and provided a brief description of each cuisine category in **lines 195-204** of the manuscript_R1: **"Specifically, home-style cuisine refers to simple everyday meals. Chinese fast food and snacks offer quick and convenient meals like noodles, dumplings, and buns. Sichuan cuisine and Hunan cuisine, both known for their spiciness and heavy flavors, were combined into one category, and in fact, many restaurants serve both cuisines. Cantonese cuisine and Fujian cuisine, characterized by their light and fresh taste, with a common use of seafood, are also merged into one category. Jiangsu cuisine and Zhejiang cuisine are both renowned for their rich and slightly sweet flavors and thus merged into one category. Other Chinese cuisine includes other local specialties in China such as Shandong cuisine and Anhui cuisine, and non-Chinese cuisine includes cuisines from countries outside China. Although these two broad categories comprise many diverse sub-cuisines, the fraction of these cuisines in the total number of restaurants in China is relatively low, so we utilize these two broad categories for classification. Notably, we've excluded catering services without fume emissions, such as tea houses and coffee houses."**

**Reference:**

Chen, C., Zhao, Y., and Zhao, B.: Emission Rates of Multiple Air Pollutants Generated from Chinese Residential Cooking, Environ. Sci. Technol., 52, 1081–1087, https://doi.org/10.1021/acs.est.7b05600, 2018.

Liang, X., Chen, L., Liu, M., Lu, Q., Lu, H., Gao, B., Zhao, W., Sun, X., Xu, J., and Ye, D.: Carbonyls from commercial, canteen and residential cooking activities as crucial components of VOC emissions in China, Sci. Total Environ., 846, 157317, https://doi.org/10.1016/j.scitotenv.2022.157317, 2022.

Lin, P., Gao, J., Xu, Y., Schauer, J. J., Wang, J., He, W., and Nie, L.: Enhanced commercial cooking inventories from the city scale through normalized emission factor dataset and big data, Environ. Pollut., 315, 120320, https://doi.org/10.1016/j.envpol.2022.120320, 2022.

Zhao, Y., Chen, C., and Zhao, B.: Is oil temperature a key factor influencing air pollutant emissions from Chinese cooking?, Atmospheric Environment, 193, 190–197, https://doi.org/10.1016/j.atmosenv.2018.09.012, 2018.

Zhao, Y. and Zhao, B.: Emissions of air pollutants from Chinese cooking: A literature review, Build. Simul., 11, 977–995, https://doi.org/10.1007/s12273-018-0456-6, 2018.

**Comment 2:**

Line 165-167: The cuisine categorization seems to be a complex process, so the R code mentioned here needs to be provided for better reproduction of results and its potential application in data processing for other regions.

**Response 2:**

Thank you for your suggestion. We have added links to source code related to cuisine categorization in the methodology section of the manuscript, including the R code and its instruction documentation, with annotations for key steps in the code. We have included the following sentence in **lines 191-192** of the manuscript_R1: **"The specific classification method is described in Text S1, and the code for cuisine categorization can be accessed at https://github.com/lizeqi18/count_cooking_emission."**

**Comment 3:**

Line 272-273: The authors need to provide a clearer description of the collection of "key policy milestones and implementation transition periods". For example, where to find comprehensive policy documents and how they relate to the control of cooking sources.

**Response 3:**

Thank you for your valuable suggestion. We have taken your suggestion into account and provided a clearer description of the collection of "key policy milestones and implementation transition periods" in Text S3 in the supplement_R1. To address this, we have included the following paragraph in Text S3 (see **lines 46-66**):

**"Text S3. Collating key policy milestones and implementation transition periods of catering pollution control policies to determine the level of control stringency**

**We obtained air pollution prevention and control policy documents and local standards for air pollution from the official websites of provincial governments or their departments of ecological and environmental protection. In response to China's Action Plan for the Prevention and Control of Air Pollutants (CPGPRC, 2013) and the Three-Year Action Plan to Win the Battle against the Blue Sky (CPGPRC, 2018), each province successively issued its provincial ordinances, action plans, or implementation plans for air pollution prevention. These documents are regularly revised and serve as instructive guidelines for air quality management within their respective provinces. They set control requirements for key**

**pollutant emission sectors, including the catering industry. The stringency of cooking emission controls specified in provincial policies has varied over different periods. Specific regulations include encouraging certain restaurants (e.g., large-scale restaurants or barbecue establishments) to install fume purifiers, strengthening pollution control in the catering industry, and comprehensive management of restaurant fume emissions. Due to China's Emission Standards for Catering Fumes (MEE, 2001) and China's Action Plan for the Prevention and Control of Air Pollutants (CPGPRC, 2013), all provinces already had some control measures for certain areas or certain restaurants. Thus, the control stringency in each province had reached level C in Table 1 before 2015. Then, we check each version of the policy document for each province to identify the year in which comprehensive control of restaurant emissions was incorporated into policies, thus determining the year each province achieved level B stringency. Additionally, a few provinces, such as Hainan and Shanghai, have issued local standards for restaurant pollutant emissions. Their local standards or policies specify the target year for all restaurants to install purification facilities, enabling us to ascertain the year these provinces reached level A stringency. Transition periods between levels can also be found within these policy documents or standards, or inferred based on assumptions in Table 1."**

Additionally, in **line 308** of the manuscript_R1, we have marked the relevant sections with **"(see Text S3 for details)"** to guide readers to the supplementary material for more comprehensive information on the collection of policy milestones and transition periods.

**Reference:**
CPGPRC (The Central People's Government of the People's Republic of China): Action plan for the prevention and control of air pollutants, http://www.gov.cn/zwgk/2013-09/12/content_2486773.htm (last access: 21 September 2023), 2013.
CPGPRC (The Central People's Government of the People's Republic of China): Three-year action plan to win the battle against the blue sky, http://www.gov.cn/zhengce/content/2018-07/03/content_5303158.htm (last access: 21 September 2023), 2018.
MEE (Ministry of Ecological Environment): Emission standards of catering oil fume (GB 13223-2003), 2001.

**Comment 4:**

Line 456-458: The spatial distribution of emissions by sector needs to be given in the supplement and briefly analyzed.

**Response 4:**

Thank you for your suggestion. We have added Figure S3 in **lines 82-84** of the supplement_R1, as shown below:

[Figure]

**Figure S3:** National gridded cooking subsector emissions in the four volatility ranges in 2021. From left to right, the sub-sectors are commercial cooking, residential cooking, and canteen cooking.

This figure displays the spatial distribution of emissions from these three cooking subsectors in the four volatility ranges.

Furthermore, we have also provided a brief analysis in **lines 506-513** of the manuscript_R1: **"Fig. S3 displays the spatial distribution of emissions from three cooking subsectors in the four volatility ranges. Commercial cooking emissions are more concentrated in economically**

**developed regions, such as provincial capitals, while less developed regions have lower emission intensity. In contrast, residential cooking emissions are correlated with population density and are distributed across areas where people live. The greater the population density, the larger the emissions. The difference between the spatial distribution of emissions from these two main cooking subsectors aligns with our understanding that in economically more developed areas, where people's disposable income is higher, people tend to dine out more frequently in commercial restaurants. Besides, canteen cooking emissions are much lower and also highly correlated with population distribution."**

**Comment 5:**

Lastly, since this paper contains a very large number of methods and data, adding a summary figure that includes the calculated data and results will help the reader to quickly spot the major methodologies and the significant values of the emission inventory.

**Response 5:**

Thank you for your constructive suggestion. We have added a summary figure "Figure 1" at the beginning of the "2 Methodology and data" section in **lines 121-126** of the manuscript_R1, as shown below:

[Figure]

**Figure 1:** Schematic of the method and data for developing a high-resolution cooking emission inventory. Three cooking emission sectors are considered. The arrows detail the emission quantification and spatial distribution methods and data used for each sector. The color of each data corresponds to the data sources of the same color indicated above. The right end of the arrows describes the calculated emission outputs with varying levels of precision.

This figure serves as a schematic representation of the methods and data utilized to develop our cooking emission inventory. We also mention this figure in **line 114** of the manuscript_R1:"We use different calculation methods for the three sources according to their characteristics and data availability, **as shown in Figure 1.**"

We believe that this addition will significantly aid readers in quickly assimilating the core methodologies and major findings of our research.

**Comment 6:**

Line 134: The full volatility range should cover the range of saturation vapor concentrations $<10^{-2}$ and $>10^7$, as indicated in Table 2, which is not expressed accurately enough here.

**Response 6:**

Thank you for pointing out this inaccurate detail. We have made necessary revisions in **lines 145-148** of the manuscript_R1 to improve the accuracy of the expression. The revised sentence now is: **"$v$ represents the volatility bin, where each bin corresponds to a range of saturation vapor concentrations (C*) of organic compounds at 300K, as defined by Chang et al. (2022). The lowest volatility bin represents the range where $\log_{10}C^* \leq -2$, and the highest volatility bin represents the range where $\log_{10}C^* > 7$. "**

**Reference:**

Chang, X., Zhao, B., Zheng, H., Wang, S., Cai, S., Guo, F., Gui, P., Huang, G., Wu, D., Han, L., Xing, J., Man, H., Hu, R., Liang, C., Xu, Q., Qiu, X., Ding, D., Liu, K., Han, R., Robinson, A. L., and Donahue, N. M.: Full-volatility emission framework corrects missing and underestimated secondary organic aerosol sources, One Earth, 5, 403–412, https://doi.org/10.1016/j.oneear.2022.03.015, 2022.

**Comment 7:**

Line 60: "to quantify" instead of "to quantifying".

**Response 7:**

Thank you for pointing out the error. We have made the necessary correction, and it now reads **"to quantify"** instead of **"to quantifying."**

---

## Author Comment (AC2)

**Dear editor and the anonymous reviewers,**

Thanks a lot for your work and time on our manuscript.

The paper entitled *"High-resolution emission inventory of full-volatility organic compounds from cooking in China during 2015-2021"* (**Manuscript ID: essd-2023-278**) by Zeqi Li, et al., has been revised carefully according to the correction requests and review reports.

The authors have addressed all the reviews' comments point-by-point as below. All the corrections and responses have been incorporated into the revised manuscript and supplement (marked with **BLUE COLORED FONTS**).

If further responses and corrections should be made, please don't hesitate to let us know.

Sincerely,

Bin Zhao, Assistant Professor
School of Environment
Tsinghua University, Beijing 100084, China
Email: bzhao@mail.tsinghua.edu.cn

**Reply on Referee #2:**

Dear reviewer,

Thank you very much for your recognition and the valuable suggestions! We have addressed the comments point-by-point as below.

All the corrections have been incorporated into the revised manuscript (manuscript_R1) and the revised supplement (supplement_R1). The point-to-point responses are listed as followed. If further responses and corrections should be made, please don't hesitate to let us know.

**Comment 1:**

Line 23: Specify the range of xLVOC (13.1 kt/yr)

**Response 1:**

Thank you for your suggestion. We have updated the information and it now reads: **"13.1 (7.36-21.0) kt/yr."**

**Comment 2:**

Introduction: Consider mentioning the popularity of each cuisine that is included in the analysis to help readers understand the results

**Response 2:**

Thank you for your constructive suggestion. In response, we have included a brief mention of two of the most popular Chinese cuisines in the introduction in **lines 55-57** of the manuscript_R1: **"With the vast regional variation, various popular Chinese cuisines such as Sichuan cuisine and Cantonese cuisine have flourished, each having distinct cooking methods and ingredients."**

Furthermore, to give readers a comprehensive understanding of the popularity of each cuisine type, we have provided a detailed description in the 'Results and Discussion' section in **lines 390-393** of the manuscript_R1: **"Fig. S2 illustrates the proportion of each cuisine in each province and across China during 2015-2021. It reveals that Chinese fast food and snacks (28.3%), home-style cuisine (20.7%) and Sichuan-Hunan cuisine (11.1%) are the most popular cuisines in China, while non-Chinese cuisine (3.21%) and barbecue (4.67%) are the least common."**

We believe these additions will help readers grasp the popularity of each cuisine and better understand the results.

**Comment 3:**

Lines 205-222: Has the uncertainty of this approach been included in section 2.4 analysis?

**Response 3:**

Thank you for your question. Yes, the uncertainty resulting from the estimation method for EFs has been incorporated in Section 2.4. Although a brief mention was initially made, in response to your feedback and for clarity, we have updated the original statement in **lines 327-332** of the manuscript_R1 as follows: **"The EFs are assumed to fit a log-normal distribution, with the CV values based on Chang et al. (2022). Since we made some estimations on the raw data of measured EFs, including using VOC or POA EFs to infer the gaseous and particle-phase full-volatility EFs, and using $PM_{2.5}$ EFs to infer POA emission factors, we also considered the additional uncertainty introduced by these estimates. For the former estimation, we added an additional 30% to the original range of uncertainty of the EFs, for instance, increasing 50% to 80%. For the latter estimation, we added an extra 20% to the original range. "** We believe that these clarifications better describe the consideration of uncertainty in our method.

Reference:

Chang, X., Zhao, B., Zheng, H., Wang, S., Cai, S., Guo, F., Gui, P., Huang, G., Wu, D., Han, L., Xing, J., Man, H., Hu, R., Liang, C., Xu, Q., Qiu, X., Ding, D., Liu, K., Han, R., Robinson, A. L., and Donahue, N. M.: Full-volatility emission framework corrects missing and underestimated secondary organic aerosol sources, One Earth, 5, 403–412, https://doi.org/10.1016/j.oneear.2022.03.015, 2022.

**Comment 4:**

Figure 3: For home-style cuisine subplot, there seems no data for Hunan and some east coastal provinces, what is the reason? How will this contribute to uncertainty?

**Response 4:**

Thank you for your inquiry regarding Figure 3, which does need some clarification. The absence

of data for certain provinces in the home-style cuisine subplot is a result of one step in our cuisine categorization method, as elaborated in the Text S1 in the supplement: "Moreover, some restaurants in certain regions have distinctive regional characteristics, such as those located in the provinces of China's eight major cuisines (Shandong, Hunan, Sichuan, Guangdong, Fujian, Jiangsu, Zhejiang, Anhui) and Xinjiang. Therefore, we also classified all homestyle restaurants in these areas as local specialty cuisines."

To enhance the explanation of Figure 3 (which has been updated to Figure 4 in the manuscript_R1) in the main text for better reader comprehension, we have slightly updated our previous description in **lines 411-413** of the manuscript_R1: "Restaurants that serve home-style dishes typically offer common, simple, traditional Chinese meals, similar to those prepared in home kitchens, as opposed to the refined and complex dishes found in larger upscale restaurants. They are not distributed in some provinces, **such as Sichuan, Hunan, and Guangdong**, because home-style restaurants in these provinces have been classified into local specialty cuisines due to their distinctive local characteristics **(specific categorization principles are available in Text S1)."**

Besides, in response to your question about the uncertainty, we have added a brief description of this cuisine classification step in **lines 182-192** of the manuscript_R1 and discussed the potential uncertainty it might introduce: **"The remaining restaurants without any specific terms in their names are categorized as home-style cuisine. However, in some provinces, such as Hunan and Guangdong, the home-style restaurants are expected to have distinct regional characteristics and are thus classified as local specialty cuisines. To explore the uncertainty introduced by this categorization method of these restaurants, we also calculated the emissions under the scenario where these restaurants remain classified as home-style cuisine instead of local specialty cuisines. Under this scenario, due to the lowest EF of home-style cuisine, the total cooking organic emissions of the involved 9 provinces (see Text S1) would decrease by 8.61% to 30.4%, and the national total cooking organic emissions would decrease by 12.2%. However, in reality, the EFs of these restaurants are probably closer to those of local specialty cuisines rather than home-style cuisine, so the actual deviation would be much less than these values."**

**Comment 5:**

Line 538: provide range for emissions

**Response 5:**

Thank you for your suggestion. We have updated the information and it now reads: **"The emissions of xLVOC, SVOC, IVOC and VOC from cooking in China in 2021 were 13.1 (7.36-21.0) kt/yr, 176 (95.8-290) kt/yr, 241 (135-374) kt/yr, and 561 (317-891) kt/yr, respectively. "**